# The UHPLC-QTOF-MS Phenolic Profiling and Activity of *Cydonia oblonga* Mill. Reveals a Promising Nutraceutical Potential

**DOI:** 10.3390/foods10061230

**Published:** 2021-05-28

**Authors:** Leilei Zhang, Gabriele Rocchetti, Gökhan Zengin, Gunes Ak, Fatema R. Saber, Domenico Montesano, Luigi Lucini

**Affiliations:** 1Department for Sustainable Food Process, Università Cattolica del Sacro Cuore, Via Emilia Parmense 84, 29122 Piacenza, Italy; leilei.zhang@unicatt.it (L.Z.); luigi.lucini@unicatt.it (L.L.); 2Department of Biology, Science Faculty, Selcuk University, Campus, Konya 42130, Turkey; gokhanzengin@selcuk.edu.tr (G.Z.); akguneselcuk@gmail.com (G.A.); 3Pharmacognosy Department, Faculty of Pharmacy, Cairo University, Kasr el-Aini Street, Cairo 11562, Egypt; fatema.saber@pharma.cu.edu.eg; 4Department of Pharmacy, University Federico II of Naples, Via D. Montesano 49, 80131 Naples, Italy

**Keywords:** *Cydonia oblonga*, nutraceuticals, polyphenols, foodomics, antioxidants, enzyme inhibitions

## Abstract

*Cydonia oblonga* Mill., normally known as the quince fruit, has been widely used in agro-food industries mainly to produce jams and jellies. However, other parts of the plants are still underutilized and not completely assessed for their nutraceutical profile. Therefore, in this work, the polyphenolic profile of *C. oblonga* was investigated using an untargeted metabolomics approach based on high-resolution mass spectrometry. Several compounds were identified in the different parts of the plants, including flavonoids (i.e., anthocyanins, flavones, flavan-3-ols, and flavonols), phenolic acids (both hydroxycinnamics and hydroxybenzoics), low-molecular-weight phenolics (tyrosol equivalents), lignans, and stilbenes. Overall, *C. oblonga* leaves showed the highest in vitro antioxidant potential, as revealed by 2,2-difenil-1-picrylhydrazyl (DPPH), 2,2′-Azino-bis(3-ethylbenzthiazoline-6-sulfonic acid) (ABTS), ferric reducing antioxidant power (FRAP), and cupric ion reducing antioxidant capacity (CUPRAC) assays, being 189.5, 285.6, 158.9, and 348.8 mg Trolox Equivalent/g, respectively. The enzymes acetyl- and butyryl-cholinesterases were both inhibited by the different plant parts of *C. oblonga*, with stems showing the higher inhibitory potential. Interestingly, the fruit extracts were the only parts inhibiting the α-glucosidase, with a value of 1.36 mmol acarbose equivalents (ACAE)/g. On the other hand, strong tyrosinase inhibition was found for stems and leaves, being 72.11 and 68.32 mg Kojic acid Equivalent/g, respectively. Finally, a high number of significant (0.05 < *p* < 0.01) correlations were outlined between phenolics (mainly anthocyanins, flava-3-ols, and tyrosol equivalents) and the different biological assays. Taken together, our findings suggest a potential exploitation of *C. oblonga* leaves and stems for the food, pharmaceutical, and cosmetic industries.

## 1. Introduction

*Cydonia oblonga* Mill. (quince) belongs to the family of Rosaceae and it is very popular in terms of nutraceutical, pharmaceutical, and ornamental properties [1]. The plant is widely distributed and cultivated in Asia, Europe, and the Middle East [2]. Generally, the quince fruit is cooked or processed before eating and is commonly used in the preparation of marmalade, jam, liqueur, and for aromatic distillation [1].

Interestingly, the different parts of this plant have been reported to possess different uses. For example, the fruits of *C. oblonga* represent an important source of pectin in the food industry [3]. In addition, the fruits contain important biologically active compounds, such as vitamin C, polyphenols, and terpenoids [4,5,6]. Moreover, the fruits have been traditionally used as drugs to treat sore throat, constipation, and bronchitis [7,8]. In addition to fruits, *C. oblonga* leaves have been studied because of their important constituents, reported to be effective against several ailments, such as diabetes, hyperlipidemia, and cancer [9,10,11]. Regarding *C. oblonga* seeds, a previous study reported a strong anti-diarrheal activity, owing to the tannin contents [1]. Recently, Sut and co-authors [12] preliminarily evaluated the quince fruit as a source of antioxidant phytoconstituents for nutraceutical and functional food applications, demonstrating that the procyanidins and caffeoyl esters of phenolic acids were the main contributors to the antioxidant activity detected.

Nevertheless, to the best of our knowledge, comprehensive information regarding the phenolic profile of the different parts of *C. oblonga* are still missing in the scientific literature. Therefore, the current study has been focused to determine the variation of chemical profiles and biological properties of *C. oblonga* fruits, leaves, and stems collected from Egypt. For the evaluation of the different biological properties, in vitro antioxidant (radical scavenging, reducing power, and metal chelating) and enzyme inhibitory (against cholinesterases, tyrosinase, α-amylase, and α-glucosidase) properties were used. Regarding chemical profiles, the extracts were analyzed as an untargeted approach via ultra-high-pressure liquid chromatography coupled with quadrupole-time-of-flight (UHPLC-QTOF) mass spectrometry. The obtained results could be useful to determine which part of the plant is characterized by the highest bioactivity for the pharmaceutical and food industries.

## 2. Materials and Methods

### 2.1. Plant Material

In this work, *C. oblonga* Mill. was collected in May 2019 from the Medicinal and Aromatic Plants Experimental Station, Faculty of Pharmacy, Cairo University, Giza, Egypt. The plant was kindly confirmed by Ms. Therese Labib, Botanical Specialist and consultant at Orman and Qubba Botanical Gardens. A voucher specimen (1.5.2019.1) was kept at the Herbarium of Pharmacognosy Department, Faculty of Pharmacy, Cairo University. The leaves, stems, and fruits of *C. oblonga* were collected from four different trees. The leaves and stems of *C. oblonga* were dried in the shade, and reduced to coarse powder using a mill. The fruits were frozen in liquid nitrogen, then kept at −80 °C until further analysis. Prior to extraction, the fruits were ground using pestle and mortar.

### 2.2. Extraction Procedure

In this work, an ultrasonic-assisted extraction was used on the ground plant material (100 g each) of the different parts of *C. oblonga,* using as extraction solvent of 1 L of methanol analytical grade (Sigma-Aldrich, Taufkirchen, Germany) at 50 °C for 45 min. The extraction step of the plant materials was done considering three replications (n = 3). Extracts were then evaporated until dry at 50 °C using a Rotary evaporator; Rotavapor^®^ R-100 (Büchi, G. Flawil, Switzerland).

### 2.3. Profiling of C. oblonga Parts by UHPLC-QTOF Mass Spectrometry

The comprehensive screening of polyphenols in the different parts of *C. oblonga* (whole fruit, stem, and roots) was done in MS full scan mode through UHPLC-QTOF mass spectrometry (Agilent Technologies, Santa Clara, CA, USA), according to a previously optimized method [13]. The dry matter of different parts of *C. oblonga* plant (20 mg) were then resuspended in 2 mL of 0.1% formic acid in 80% methanol and filtered with 0.2 μm cellulose membrane into vials for UHPLC-QTOF-MS. The samples were analyzed in triplicate, considering an injection volume of 6 μL. Overall, a randomized pattern was adopted for the injection sequence, alternating every 10 injections with quality control (QC) samples analyzed in data-dependent MS-tandem as previously described [13]. The QC samples were made by pooling an aliquot of each extract. The raw MS data were processed using an Agilent Profinder (Agilent Technologies) software (version B.06) by a ‘find-by-formula’ algorithm, according to the parameters previously described [13]. In this regard, the Phenol-Explorer (version 3.6) database was used as reference for putative annotation, based on level 2 of confidence (i.e., putative identification exploiting the isotopic profile of each compound and a mass accuracy < 5 ppm). In addition, in order to get a higher degree of confidence, a further identification/confirmation step was carried out using MS-DIAL software (version 4.24) [14] to annotate compounds in the QCs and using publicly available MS/MS libraries (e.g., Mass Bank of North America; MoNA) and MS-Finder software (version 3.46) [15] to identify the compounds according to an in silico fragmentation approach, and using the Lipid Maps, FoodDB, and PlantCyc libraries as previously described [16].

Finally, the major classes of polyphenols were quantified using standard compound solutions analysed with the same MS conditions. The pure compounds chosen as representatives of their main class were: cyanidin (anthocyanins), luteolin (flavones and other flavonoids), quercetin (flavonols), catechin (flavan-3-ols), sesamin (lignans), tyrosol (low-molecular-weight phenolics), ferulic acid (phenolic acids), and resveratrol (stilbenes). Compounds were provided by Extrasynthese (Lyon, France) and they were characterized by a purity > 98%. Linear regression curves were built to quantify the main representative subclasses obtained (R^2^ values > 0.98; using the following five concentrations: 0.1, 1, 10, 100, and 1000 mg/L), and the semi-quantitative values were then expressed as mg equivalents/g dry matter (DM).

### 2.4. In Vitro Antioxidant Potential and Enzymes-Inhibitory Activities

The *C. oblonga* methanolic extracts were evaluated for their in vitro antioxidant activity, using different assays such as DPPH, ABTS, CUPRAC, FRAP, phosphomolybdenum, and metal chelating. The exhaustive steps regarding each method are previous described [17]. To evaluate the results from the antioxidant assays, we used equivalents of standards (Trolox (TE)/g for radical scavenging and reducing the power assays and the EDTA (EDTAE) for the metal chelating assay). To evaluate the inhibitory effects of the tested extracts on some enzymes (α-amylase, α-glucosidase, tyrosinase, and cholinesterases), spectrophotometric methods were used. Similar to antioxidant assays, a standard equivalent way was used to evaluate the results. Some compounds (acarbose (ACAE), galantamine (GALAE), and kojic acid (KAE)) were used as standard inhibitors. The different *C. oblonga* extracts were analyzed in triplicate (n = 3).

### 2.5. Statistics and Chemometrics

The analysis of variance (one-way ANOVA; *p* < 0.05) with Duncan’s post hoc test was performed using PASW Statistics 26.0 (SPSS Inc., Chicago, IL, USA) to investigate the significant differences in semiquantitative values on each subclass of polyphenols, in vitro antioxidants, and in vitro enzymatic inhibitory properties for the different parts of *C. oblonga*. In addition, Pearson’s correlation coefficients (*p* < 0.01 and *p* < 0.05; two-tailed) were obtained to find out the significant correlations between phytochemical contents and biological activities (PASW Statistics 26.0, SPSS Inc., Chicago, IL, USA).

Regarding the metabolomics-based dataset, the data filtering and normalization process was done with the Agilent Mass Profiler Professional software (from Agilent Technologies, Santa Clara, CA, USA; version B.05.00) as previously described [18]. An unsupervised hierarchical cluster analysis (HCA) was carried out, setting the similarity measure as ‘Euclidean’ and ‘Wards’ as the linkage rule. Afterward, the raw dataset was then interpreted through orthogonal projection to latent structures discriminant analysis (OPLS-DA) supervised analysis using SIMCA 16 (Umetrics, Malmo, Sweden). Cross-validated ANOVA (*p* < 0.01) and permutation testing (N = 200) were used for model validation and to exclude overfitting, respectively. Moreover, the OPLS-DA model was investigated for the fitness parameters (goodness-of-fit R^2^Y and goodness-of-prediction Q^2^Y) and outliers, according to the Hoteling’s T2 test (95% and 99% confidence limit for suspect and strong outliers, respectively). The variable importance in projection (VIP) was adopted to choose the most discriminant compounds of *C. oblonga* extracts, selecting those variables with the highest discrimination potentials (VIP score > 1.2), provided with fold-change values in order to provide the accumulation trend of each marker compound.

## 3. Results and Discussion

### 3.1. Phenolic Profiling of C. oblonga by UHPLC-QTOF Mass Spectrometry

A comprehensive investigation based on the phenolic profiling of the different parts of *C. oblonga* was done using an untargeted metabolomic approach. This method enabled the putative identification of 275 compounds (excluding the potential isomeric forms of phenolics), thus demonstrating a complex phenolic profile, majorly represented by 149 flavonoids (i.e., anthocyanins, flavones, flavanols, flavonols), 58 phenolic acids (i.e., hydroxycinnamic and hydroxybenzoic acids), 42 low-molecular-weight phenolics (i.e., tyrosols, phenolic terpenes, hydroxycoumarins, furanocoumarins, hydroxybenzaldehydes), 18 lignans, and 8 stilbenes. A full list of detected compounds, collectively with their abundance and mass spectra, is available in the Appendix A. In addition, using the tandem-mass spectrometry approach, the structural identity of some phytochemicals was confirmed (Appendix A). In particular, several flavonoids, with their glycosides such as cyanidin, delphinidin, procyanidin B1, epicatechin, kaempferol, followed by main important lignans, phenolic acids and stilbenes compounds, have been detected and confirmed. In addition, the tandem-MS/MS method was also used to confirm the structural identity of 52 compounds, mainly belonging to flavonoids, lignans, and phenolic acids. Moreover, some lipid compounds, such as linoleic acid derivatives and triterpenoids, were detected. All the compounds annotated via MS/MS are available in Appendix A, together with their ontology, relative abundances, and other annotation-related parameters.

Thereafter, the quantification per class of the different phytochemicals was performed using representative standard compounds, and the results are reported in Figure 1 and Appendix A.

Generally, the total phenolic content (TPC) ranged from 158.7 mg·Eq./g (for fruits) up to 262.9 mg·Eq./g (for stems). The flavonoid content in leaves was almost double the amount found in stems and whole fruits (78.1, 44.9, and 27.3 mg·Eq./g, respectively) as shown in Appendix A. The annotated flavonoid glycosides in *C. oblonga* extracts include both *O*- and *C*-glycosides, which are found in higher abundance in the leaves and stems. These are mainly represented by apigenin 7-*O*-glucoside, naringenin-*O*-glucoside, chrysoeriol 7-*O*-(6″-malonyl-apiosyl-glucoside), dihydroquercetin 3-*O*-rhamnoside, apigenin 6-*C*-glucoside, luteolin 6-*C*-glucoside, and apigenin 6,8-di-*C*-glucoside. On the other hand, phenolic acids, especially hydroxycinnamic acids, constitute an integral part of *C. oblonga*’s phytochemical profile (Appendix A). In particular, the most abundant compounds characterizing this class were found to be 5-caffeoylquinic/3-caffeoylquinic acids, *p*-coumaric acid 4-*O*-glucoside, ferulic acid 4-*O*-glucoside feruloylquinic acid, sinapoylquinic acid, 3,5-dicaffeoylquinic acid, and 3,5-diferuloylquinic acid. The presence of flavonoid-*C*-glycosides, procyanidin dimers, and trimers in addition to caffeoylquinic acids and derivatives in *C. oblonga* extracts (Appendix A) could be responsible for its bioactive properties, as previously suggested in the literature [1,19,20]. Interestingly, dihydrochalcones, namely phloridzin, phloretin, phloretin 2′-*O*-xylosyl-glucoside, and 3-hydroxyphloretin 2′-*O*-xylosyl-glucoside, were detected in all investigated plant parts, and were relatively higher in *C. oblonga* leaves. Nevertheless, Wojdyło et al. [21], described the absence of dihydrochalcones in *C. oblonga* fruits in contrast to apples. On the other hand, lignans were found in appreciably high amounts in Cydonia stems, amounting to 127.7 mg Eq./g (leaves and fruits). Previous studies reported the isolation and characterization of lignans from Family Rosaceae [22,23]. Therefore, this work can be considered as the first reporting on their presence in *C. oblonga* plant parts.

### 3.2. Multivariate Discrimination of the Different Plant Parts

Multivariate statistical analyses, namely both supervised and unsupervised approaches, were exploited to investigate those marker compounds most involved in the discrimination of *C. oblonga* extracts. The heat-map HCA (produced from the fold-change of each annotated compound in the different extracts) was used to group the plant extracts according to intrinsic similarities in their chemical profiles (Figure 2).

As can be observed, the HCA showed two main clusters and three distinct subclusters. The leaves and stems were found to have a close similarity, while the whole fruits constituted another main cluster. Accordingly, as reported in Figure 1, the whole fruit showed the lowest content of polyphenols when compared to the other parts. On the other hand, the similarity of the stems and leaves was confirmed by both the amount and the profile of the phytochemicals annotated.

Afterward, in order to better distinguish the differences/similarities in the phenolic profiles across the different *C. oblonga* extracts, a supervised OPLS-DA model was created followed by the selection of the main VIP maker compounds. The resulting OPLS-DA score plot showed distinct sample segregation by introducing an orthogonal signal correction (Figure 3). Particularly, the orthogonal component showed the ability to discriminate the leaf matrix from the stem and whole fruit matrices of *C. oblonga*. This result suggests that the phytochemical profile of the whole fruit and the stem are more similar in terms of phytochemical composition. Moreover, the latent vector t [2] showed the ability to discriminate the whole fruit matrix from the stem and leaf matrices. Interestingly, the second latent vector provided a discrimination that was in strict accordance with the semi-quantitative values reported in both Appendix A and Figure 1.

The model obtained was also characterized by strong parameters with R^2^Y = 0.998 (goodness-of-fit) and Q^2^Y = 0.976 (goodness-of-prediction). Indeed, the prediction model successfully demonstrated the impact of several phenolic compounds for discrimination purposes. The VIP compounds (VIP score > 1.2) were exported and listed in Table 1, provided with their chemical class, VIP score ± standard error, and a Log2 Fold Change (LogFC) value for the pairwise comparison between the different plant extracts (i.e., leaf vs. stem, leaf vs. whole fruit, and stem vs. whole fruit). The VIP scores displayed the overall importance of each variable (x) on all responses (Y) cumulatively over all components. It is a weighted sum of squares of the OPLS weights, taking into account the amount of explained Y-variance in each dimension. The VIP values reflect the importance of the terms in the model both with respect to Y, i.e., its correlation to all the responses, and with respect to X (the projection). In addition, the VIP score in the OPLS model is the combination of three vectors, namely VIP for the predictive components (VIP predictive), VIP for the orthogonal components (VIP orthogonal), and VIP for the total model (VIP total).

Overall, the fold-change analysis was useful to reveal specific marker compounds of the non-edible parts of the plant (i.e., the leaves and stems), thus enlarging the knowledge on their potential health-promoting properties. Overall, 29 compounds were the most discriminating phenolic markers in *C. oblonga* organs, characterized by a higher prevalence of flavonoids (e.g., anthocyanins, dihydrochalcones, flavanols, flavonols, and flavones) followed by phenolic acids and other polyphenols in smaller numbers. Regarding the anthocyanins subclass, the most discriminant markers were petunidin 3-*O*-glucoside, delphinidin 3-*O*-arabinoside, cyanidin 3-*O*-xyloside, cyanidin 3-*O*-xylosyl-rutinoside, and pelargonidin 3-*O*-sambubioside, and were most up-accumulated in the stem and leaf extracts (with cumulative LogFC values of 43.35 and 19.54; respectively) compared to the whole fruits. In addition, for the remaining flavonoids markers (such as 3-hydroxyphloretin 2′-*O*-xylosyl-glucoside, dihydroquercetin 3-*O*-rhamnoside, theaflavin, eriodictyol 7-*O*-glucoside, luteolin 7-*O*-glucuronide, quercetin 3-*O*-xylosyl-rutinoside, kaempferol 3-*O*-glucuronide, and others), stem extracts resulted as the best source of phytochemicals (cumulative LogFC 90.41 and 92.78; whole fruit and leaf, respectively) (Table 1). Regarding the class of phenolic acids, the hydroxycinnamic acids were the major subclass of phenolic compounds found as VIP discriminant markers, including *p*-coumaroylquinic acid, stigmastanol ferulate, and 3,5-dicaffeoylquinic acid, and were distributed mostly in the stem and leaf extracts (cumulative LogFC values of 25.11 and 14.21, respectively). The supervised OPLS-DA model was able to discriminate only those hydroxycinnamic acids which are more represented in the leaves and stems than in the fruits. The hydroxycinnamic acids are widely distributed in plants and are detected at high concentrations in fruits and vegetables [24]. Stojanović et al. [25] reported similar hydroxycinnamic acid profiles in the peel and pulp of *C. oblonga* Mill. (when considering caffeoylquinic acid or dicaffeoylquinic acid, 4-*p*-coumaroylquinic acid and derivatives). No available data were found regarding the high concentrations of these compounds in the methanolic extracts of stems and leaves. Moreover, 3,5-dihydroxybenzoic acid/protocatechuic acid (belonging to hydroxybenzoic acids) were clearly up-accumulated in the leaves and fruits (LogFC values of 19.83 and 21.58, respectively).

### 3.3. In Vitro Antioxidant and Enzyme Inhibitory Activities

To shed light on the possible biological valorization of *C. oblonga*, the different plant extracts were evaluated for their in vitro biological properties (Table 2).

Interestingly, the leaves showed strong antioxidant and radical scavenging activity, as revealed by different antioxidant assays. The stems of *C. oblonga* also showed a comparable activity to the leaves although less active, while the fruits demonstrated weak in vitro antioxidant potential. On the other hand, all the investigated organs displayed metal chelating power with 10.80, 11.15, and 11.66 mg EDTAE/g for the leaves, fruits, and stems, respectively. The promising antioxidant potential of *C. oblonga* could be related to its high total phenolic (TPC) and total flavonoid contents (TFC). Several previous reports have been focused on the antioxidant effect of *C. oblonga* fruits, being processed in the form of jams and jellies and owing this potential to the encompassed caffeoylquinic acids and derivatives [26,27]. However, no information is actually available in thr literature on the stems’ in vitro antioxidant activity.

Overall, the target enzyme inhibitors of acetylcholine- and butyrylcholine-esterases (i.e., AChE and BChE) are important agents commonly implicated to control neurodegenerative diseases [28,29]. Our findings revealed that all studied *C. oblonga* organs exerted in vitro inhibitory action on both AChE and BChE, with the stems presenting a numerically higher inhibition, being 2.54 and 2.95 mg galantamine equivalent (GALAE)/g, respectively (Table 3). In this regard, no significant differences were detected for AChE inhibition when considering the three plant parts, whilst fruit and stem extracts showed a significantly higher (*p* < 0.05) BChE inhibition when compared with the leaves. The observed trend could be in part related to the enrichment of *C. oblonga* stems with lignans (Appendix A). In this regard, naturally occurring lignans have been previously described to have a potential inhibitory action on acetylcholinesterase [30,31]. Moreover, the presence of flavonoids, hydroxycinnamic acids, and procyanidins mostly contributed to the observed anticholinesterase activity. It is noteworthy that the mechanistic non-competitive inhibition of phenolics and flavonoids on AChE involves the interaction with its peripheral anionic sites [32].

Regarding the antidiabetic potential of *C. oblonga* extracts, the in vitro inhibitory action on the key enzymes, namely α-amylase and α-glucosidase, was assessed. The results (Table 3) showed that different organs of *C. oblonga* were active to provide the inhibition of α-amylase, with the leaves and stems presenting a numerically higher activity (although not a significant difference) when compared to the fruits (being 0.48, 0.43, and 0.29 mmol·ACAE/g, respectively). Interestingly, only fruits caused the inhibition of α-glucosidase enzyme with 1.36 mmol ACAE/g. Being supported by the traditional findings of the uses of *C. oblonga* leaves to ameliorate diabetic conditions in Turkey, Aslan and co-workers [33] investigated the in vivo antidiabetic activity of *C. oblonga* leaves. Accordingly, authors reported that the blood glucose level showed a 33% decrease in diabetic rats upon regular intake of 500 mg/kg *C. oblonga* leaf extract for 5 days. On the contrary, Nyambe-Silavwe & Williamson [34] claimed the efficacy of 5-caffeoylquinic acids (5-CQA) and other phenolic acids as inhibitors of amylase and maltase. They found only very weak inhibitory potential of these compounds when tested on enzymes from human source. Yet, the antidiabetic potential of *C. oblonga* extracts on α-amylase could be correlated to the flavonoid content and to triterpenes, the latter have not been investigated in the current study. Furthermore, in-depth mechanistic studies are to be followed to unravel the target proteins that are mostly affected by *C. oblonga* bioactives.

Regarding the other activities under investigation, tyrosinase is an important key enzyme involved in the regulation of melanogenesis. In this regard, L-tyrosine is oxidized to dopaquinone, the latter further being converted to eumelanin and pheomelanin [35]. Consequently, tyrosinase inhibitors are widely used in cosmetic preparations to control the hyperpigmentation and in skin-whitening products, and has been applicable in the agro-food industry to delay the browning of fruits and vegetables [36].

Thereby, different organs of *C. oblonga* have been evaluated for possible tyrosinase inhibitory activity. Interestingly, the stems and leaves showed a high tyrosinase inhibitory potential with 72.11 and 68.32 mg Kojic Acid Equivalent (KAE)/g, respectively. In addition, the whole fruits of *C. oblonga* exerted a less potent effect with 56.64 mg KAE/g. Previous studies described flavonoids as represented by chalcones and kaempferol in addition to hydroxystilbenes (resveratrol and derivatives) as potent tyrosinase inhibitors [37]. Further, the glycosylation of resveratrol could potentiate this inhibitory tyrosinase activity [38]. The stems of *Cydonia*, with their high content of lignans (Table 1), could also contribute to this action. Sesamin and sesamolin inhibited the melanin synthesis via tyrosinase inhibitory activity [39]. In the same context, Wu, et al. [40] characterized new lignans from *Castanea henryi* and further reported their tyrosinase inhibitory activity. Moreover, anthocyanins are well-documented for their tyrosinase inhibition activity [41,42]. Of note, no previous studies were found concerning the promising tyrosinase inhibitory action of *C. oblonga.*

### 3.4. Correlations between Phytochemical Profiles and Biological Activities

Pearson’s correlation analysis was carried out and correlation coefficients were determined to find the probable correlations among the phenolic classes and the in vitro biological activities of *C. oblonga* extracts. The significant correlation coefficients are shown in Appendix A. Overall, flavonoids (flavan-3-ols, flavones, flavonols, and anthocyanins) in addition to low-molecular-weight phenolics were positively and significantly (*p* < 0.01) correlated with different in vitro antioxidant assays (i.e., DPPH, ABTS, FRAP, CUPRAC, and phosphomolybdenum). Regarding the tyrosinase inhibition, anthocyanins and lignans exhibited the highest correlation coefficients (0.867, 0.886) followed by flavonols (0.846).

Previous reports investigated the potential of anthocyanins as tyrosinase inhibitors and also their ability to control melanogenesis process [43,44,45]. Furthermore, lignans (such as sesamol, sesamolin, and sesamin) have been previously studied as powerful tyrosinase inhibitors [39,46]. Additionally, significant correlation coefficients (*p* < 0.01) were detected among flavonoids (anthocyanins, in particular) and low molecular weight phenolics with alpha amylase inhibition. Interestingly, anthocyanin intake was previously correlated to delayed carbohydrate digestion via the inhibition of pancreatic amylase and glucosidase [47]. Overall, the correlations observed in the current study provide scientific evidence for the proper exploitation of *C. oblonga* extracts as a functional food in the management of several ailments due to its high levels of antioxidants, tyrosinase, AChE and BChE, and amylase inhibition.

## 4. Conclusions

The phenolic profile of different organs of *C. oblonga* Mill. has been investigated via untargeted metabolomic approach using UHPLC-QTOF mass spectrometry. The findings showed that *C. oblonga* extracts are a promising source of health promoting phenolics, including flavonoids (such as anthocyanins), phenolic acids, and lignans. The complex phytochemical profile of *C. oblonga* presented herein in addition to its potent in vitro antioxidant and enzyme inhibitory activities strongly makes it as a potential source for cosmetic and nutraceutical industries. In particular, the leaves and stems should be regarded as a rich source of phenolics, and more attention should be directed on their large-scale exploitation in phyto-pharmaceutical industries. Our future perspective will be targeted to optimize *C. oblonga* phenolic yield especially from leaves and stems, in context of response surface methodology (RSM).

## Figures and Tables

**Figure 1 foods-10-01230-f001:**
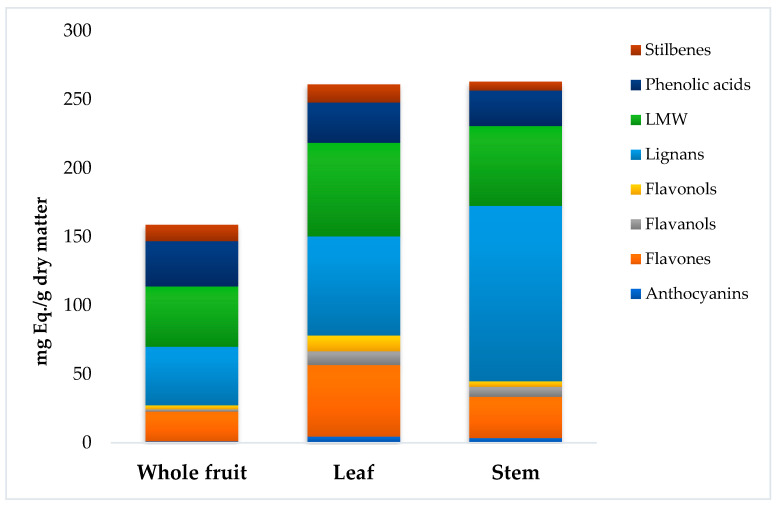
Cumulative phenolic content of *Cydonia oblonga* Mill. hydroalcoholic extracts (80% methanol), namely whole fruit, leaf, and stem. The semi-quantitative values are expressed as the mean of three replicates, mg Equivalent (Eq.)/g. LMW = lower-molecular-weight phenolics (i.e., as tyrosol equivalents).

**Figure 2 foods-10-01230-f002:**
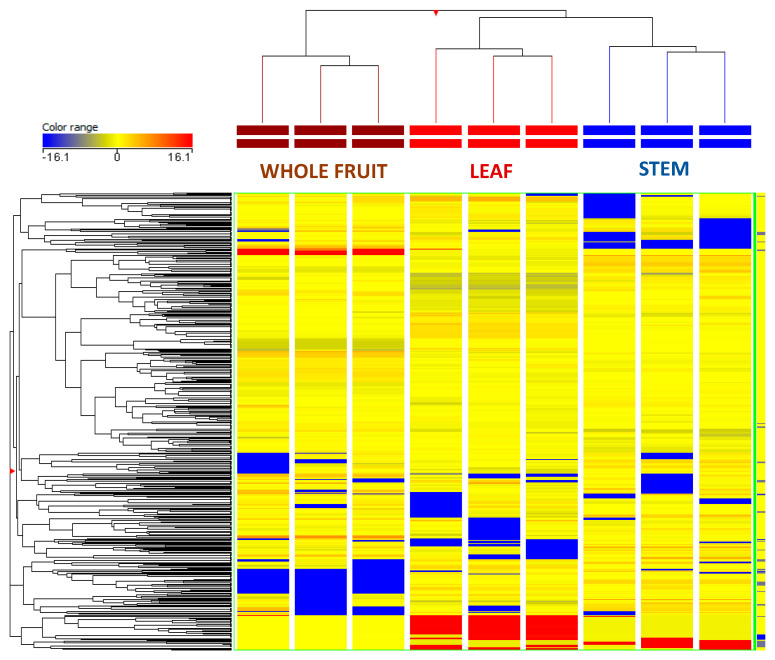
Hierarchical cluster analysis (HCA) built according to phenolic profile of *Cydonia oblonga* Mill. considering the different plant extracts. The cluster was built using Log2 median normalized values (similarity: Squared Euclidean; linkage rule: Ward). The heat-map colour range in each column represents the maximum (red) and minimum (blue) fold-change values.

**Figure 3 foods-10-01230-f003:**
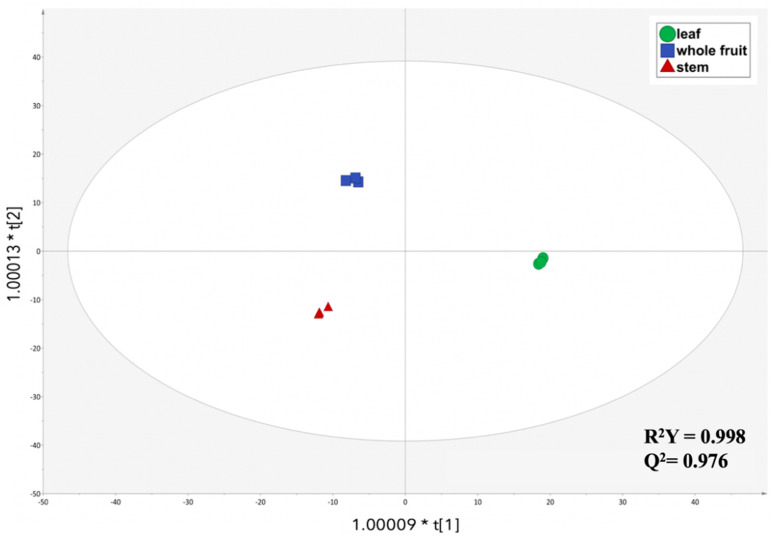
Orthogonal projections to latent structures discriminant analysis (OPLS-DA) score plot built according to phenolic profiling of *Cydonia oblonga* Mill. and considering the different plant extracts as discrimination parameter.

**Table 1 foods-10-01230-t001:** Discriminant polyphenol compounds identified by the VIP (variable importance in projection) with a score > 1.2. VIP makers were classified in classes and subclasses according to the Phenol Explorer database. Each compound as provided with a VIP score ± standard error and log fold-change values obtained by pairwise comparison among different *Cydonia oblonga* Mill. organs.

Class	Sub-Class	Discriminant Compound (OPLS-DA)	VIP Score ± SE	Log FC [Leaf] vs. [Stem]	Log FC [Leaf] vs. [Whole Fruit]	Log FC [Stem] vs. [Whole Fruit]
Flavonoids	Anthocyanins	Petunidin 3-*O*-glucoside	1.23 ± 0.31	−2.63	1.98	4.61
		Delphinidin 3-*O*-arabinoside/Delphinidin 3-*O*-xyloside	1.23 ± 0.30	−2.45	−0.08	2.37
		Cyanidin 3-*O*-xyloside/Cyanidin 3-*O*-arabinoside	1.22 ± 0.33	−2.55	−0.22	2.33
		Cyanidin 3-*O*-(6″-malonyl−3″-glucosyl-glucoside)	1.22 ± 0.32	0.68	−1.43	−2.11
		Cyanidin 3-*O*-sambubioside 5-*O*-glucoside	1.21 ± 0.35	−5.03	−2.14	2.89
		Pelargonidin 3-*O*-sambubioside	1.21 ± 0.33	−7.07	5.10	12.16
		Peonidin 3-*O*-(6″-acetyl-glucoside)	1.21 ± 0.33	−3.58	0.22	3.81
		Cyanidin 3-*O*-xylosyl-rutinoside	1.21 ± 0.27	−1.18	16.11	17.29
	Dihydrochalcones	3-Hydroxyphloretin 2′-*O*-xylosyl-glucoside	1.20 ± 0.32	−4.04	−2.99	1.05
		Dihydroquercetin 3-*O*-rhamnoside	1.21 ± 0.37	−3.72	−1.56	2.16
	Flavanols	Theaflavin	1.21 ± 0.34	−21.79	−18.84	2.96
	Flavanones	Eriodictyol	1.22 ± 0.34	−1.60	7.36	8.97
		Eriodictyol 7-*O*-glucoside	1.21 ± 0.37	−3.72	−1.56	2.16
	Flavones	Cannflavin A	1.21 ± 0.34	1.13	−1.36	−2.49
		Chrysoeriol 7-*O*-(6″-malonyl-glucoside)	1.21 ± 0.33	−7.01	−0.43	6.58
		Luteolin 7-*O*-glucuronide	1.21 ± 0.33	−21.43	−1.39	20.04
		Cirsimaritin	1.20 ± 0.27	−0.16	−2.97	−2.80
	Flavonols	Quercetin 3-*O*-xylosyl-glucuronide	1.23 ± 0.28	−1.92	16.55	18.47
		Kaempferol 3-*O*-(2″-rhamnosyl−6″-acetyl-galactoside) 7-*O*-rhamnoside	1.22 ± 0.31	−2.21	16.25	18.45
		Quercetin 3-*O*-xylosyl-rutinoside	1.21 ± 0.35	−3.42	−0.97	2.45
		Kaempferol 3-*O*-glucuronide	1.21 ± 0.33	−21.41	−1.39	20.02
		Kaempferide	1.21 ± 0.33	−10.61	−7.96	2.65
		Myricetin	1.20 ± 0.27	17.61	−4.19	−21.80
	Isoflavonoids	Glycitin	1.22 ± 0.32	−3.02	7.17	10.19
		6″-*O*-Malonylgenistin	1.20 ± 0.32	−5.45	−4.09	1.36
Phenolic acids	Hydroxybenzoic acids	Protocatechuic acid / 3,5-Dihydroxybenzoic acid	1.22 ± 0.34	19.83	−1.75	−21.58
	Hydroxycinnamic acids	*p*-Coumaroylquinic acid/3-*p*-Coumaroylquinic acid/4-*p*-Coumaroylquinic acid/5-*p*-Coumaroylquinic acid	1.22 ± 0.33	−3.97	−0.78	3.18
		Stigmastanol ferulate	1.20 ± 0.44	−2.85	1.85	19.70
		3,5-Dicaffeoylquinic acid/4,5-Dicaffeoylquinic acid/3,4-Dicaffeoylquinic acid	1.20 ± 0.33	−4.09	−1.86	2.23

**Table 2 foods-10-01230-t002:** In vitro antioxidant assays of different *Cydonia oblonga mill. extracts*.

	DPPH (mg TE/g)	ABTS (mg TE/g)	CUPRAC (mg TE/g)	FRAP (mg TE/g)	Metal Chelating mg (EDTAE/g)	Phosphomolybdenum (mmol TE/g)
Leaf	189.53 ± 0.39 ^a^	285.65 ± 4.90 ^a^	348.84 ± 1.87 ^a^	158.86 ± 2.90 ^a^	10.80 ± 0.15 ^a^	2.31 ± 0.17 ^a^
Whole fruits	6.53 ± 0.31 ^c^	9.15 ± 1.07 ^c^	24.84 ± 0.29 ^c^	13.66 ± 0.03 ^c^	11.15 ± 0.58 ^a^	0.56 ± 0.03 ^a^
Stem	129.68 ± 2.31 ^b^	174.32 ± 5.66 ^b^	215.60 ± 3.38 ^b^	120.86 ± 2.09 ^b^	11.66 ± 0.14 ^a^	1.85 ± 0.06 ^a^

Values are expressed as the mean ± Standard Deviation (n = 3) as mg·Eq/g dry matter. Different superscript letters differ in *p* < 0.05, Tukey’s post hoc. TE: trolox equivalent; EDTAE: Ethylenediaminetetraacetic acid equivalent.

**Table 3 foods-10-01230-t003:** Enzyme inhibitory activities of different *Cydonia oblonga mill. extracts*.

	AChE (mg GALAE/g)	BChE (mg GALAE/g)	Tyrosinase (mg KAE/g)	*α*-Amylase (mmol ACAE/g)	*α*-Glucosidase (mmol ACAE/g)
Leaf	2.37 ± 0.04 ^a^	2.35 ± 0.07 ^b^	68.32 ± 0.30 ^b^	0.48 ± 0.00 ^a^	n.d
Whole fruits	2.36 ± 0.02 ^a^	2.94 ± 0.13 ^a^	56.64 ± 0.12 ^c^	0.29 ± 0.03 ^a^	1.36 ± 0.09
Stem	2.54 ± 0.02 ^a^	2.95 ± 0.07 ^a^	72.11 ± 0.26 ^a^	0.43 ± 0.00 ^a^	n.d.

Values are expressed as the mean ± Standard Deviation (n = 3) as mg Eq/g dry matter. Different superscript letters differ in *p* < 0.05, Tukey’s post *hoc*. GALAE: Galantamine equivalent; KAE: Kojic acid equivalent; ACAE: Acarbose equivalent; n.d.: not detected.

## Data Availability

Not applicable.

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
