# Peer review of "The UHPLC-QTOF-MS Phenolic Profiling and Activity of Cydonia oblonga Mill. Reveals a Promising Nutraceutical Potential"

_foods, 2021, doi:10.3390/foods10061230_

Round 1

Reviewer 1 Report

The manuscript “The UHPLC-QTOF-MS phenolic profiling and activity of Cydonia oblonga Mill. reveals a promising nutraceutical potential” is interesting in terms of new information given about the content of bioactive compounds in leaf, stem, and fruit. However, it needs a lot of improvement to be published.

Details:

Abstract lines 26 and 32, and Keywords – write Cydonia oblonga instead of Cydonia. Use italics.

Introduction line 62 and Materials and Methods line 83 – define the abbreviations only once – preferably in the Introduction section.

Materials and Methods points 2.2 and 2.3 (lines 85-86) – which extraction method was used? Was it extraction with methanol or extraction with 0.1% formic acid : methanol 20 : 80? According to point 2.2 ground plant material was used? How was it ground? Was it dried and ground or ground in liquid nitrogen? How was it dried? The same questions refer to point 2.3.

Point 2.3 line 82 (and generally in entire paper) – Cydonia oblonga Mill. is a tree. What do you exactly mean by the stem?

Point 2.3 lines 109-111 There is something mentioned about linearity. However, method validation is not given in the paper. It is especially interesting because the active compounds are at very different concentration levels and QTOF linear range is not wide. The matrix effect in the biological material is also important.

Point 2.4 line 122 – what extracts? These prepared in methanol according to point 2.2? Write “ The C. oblonga methanolic extracts …”.

Point 3.1 line 157 – why some abundances were equal to 1 (especially in one out of three samples), e.g. for peralgonidin 3-O-arabinoside?.

Point 3.1 line 158 – replace trough with through.

Point 3.1 line 164 – write MS/MS.

Point 3.1 line 167 – the names of extracts in Table S2 give no information. Which extract is from leaf, stem, and fruit? This information is given only in Table S1.

Figure 1 Give “Whole fruit” as the first bar, then “Leaf” and “Stem” (like in Figure 2). Is it the methanolic extract according to point 2.2? The extract in the UHPLC method is not methanolic (only 80% methanol was used).

Point 3.1 line 180 – what is appropriate – “higher in leaves and stems than in fruit” or “higher in leaves than in stems”? I cannot verify based on the information given in Tables S1 and S2.

Figure 2 – write Cydonia oblonga in italics.

Table 1 – it should be preferably on one page – the header cannot be easily followed when it is on a separate page.

Point 3.3 lines 286-287 – based on the information given in Table 3 there is no statistically valid difference for AChE results (all plant parts) and between BChE (for fruit and stem). You cannot write in the text that these results were higher for the stem. Further discussion in the text should also be corrected.

Point 3.3 lines 302-303 – based on the information given in Table 3 there is no statistically valid difference for inhibition of amylase.

Point 3.3 lines 302-305 – this is not clear – shouldn’t you write it in one sentence?

Point 3.3 line 329 – Figure 1?

Author Response

Reviewer #1

The manuscript “The UHPLC-QTOF-MS phenolic profiling and activity of Cydonia oblonga Mill. reveals a promising nutraceutical potential” is interesting in terms of new information given about the content of bioactive compounds in leaf, stem, and fruit. However, it needs a lot of improvement to be published.

Authors: We would like to thank the reviewer for having appreciated this work. A deep revision has been done according to the major drawbacks raised by the reviewer.

Details:

Abstract lines 26 and 32, and Keywords – write Cydonia oblonga instead of Cydonia. Use italics.

Authors: revised, accordingly.

Introduction line 62 and Materials and Methods line 83 – define the abbreviations only once – preferably in the Introduction section.

Authors: revised, accordingly.

Materials and Methods points 2.2 and 2.3 (lines 85-86) – which extraction method was used? Was it extraction with methanol or extraction with 0.1% formic acid : methanol 20 : 80? According to point 2.2 ground plant material was used? How was it ground? Was it dried and ground or ground in liquid nitrogen? How was it dried? The same questions refer to point 2.3.

Authors: Leaves and stems of C. oblonga were dried in shade, and reduced to coarse powder using a mill. The fruits were frozen in liquid nitrogen, then kept at -80 °C till further analysis. Prior to extraction, the fruits were ground using pestle and mortar.

The different parts of C. oblonga plant were extracted by an ultrasound-assisted extraction (starting from 100 g of plant material), using methanol as extraction solvent and considering the following working conditions: Temperature: 50°C; Time: 45 min. Extracts were then evaporated till dryness at 50°C using Rotary evaporator; Rotavapor® R-100 (Büchi, G. Switzerland).

Afterwards, the obtained extracts were sent to Italy for the untargeted metabolomic profiling analysis. In this regard, the sample preparation for UHPLC-QTOF mass spectrometry analysis consisted of resuspending the dried extracts in a typical working solution of methanol 80% + 0.1% of formic acid (v/v), thus filtering the resuspended extracts through 0.2-micron cellulose syringe filters.

Point 2.3 line 82 (and generally in entire paper) – Cydonia oblonga Mill. is a tree. What do you exactly mean by the stem?

Yes, Cydonia oblonga Mill. is a tree. Stems are the small branches and twigs of C. oblonga carrying the leaves.

Point 2.3 lines 109-111 There is something mentioned about linearity. However, method validation is not given in the paper. It is especially interesting because the active compounds are at very different concentration levels and QTOF linear range is not wide. The matrix effect in the biological material is also important.

Authors: We understand the author's request. Please, consider that the semi-quantitative analysis described in paragraph 2.3. was previously used by our research team in more than 70 published papers on high-impact journals when considering untargeted phenolic profiling by UHPLC-QTOF mass spectrometry. Accordingly, in this work, we were in the position of deciding between using the strongest ID confidence (MS/MS) with a limited compound coverage, rather than having a lower ID confidence but a very comprehensive profiling. We decided for the second mainly for three reasons: a) less characterized and commercially unavailable compounds (e.g. diversely conjugated and glycosylated) are often very informative and surely novel; b) we added several ID criteria, additionally to monoisotopic accurate mass, to compensate for the lack of MS2 (namely, isotopic spacing, isotopic ratio, filters by frequency across replications, retention time alignment); c) For quince, compounds coverage in available datasets are even more incomplete than for other non-model plants. Also, phenolic compounds are featured by a rather wide chemical diversity. Therefore, we decided to use several surrogates for semi-quantification, because we assumed these surrogates could better represent the ESI behavior of chemically related compounds. We are aware this is a compromise (i.e., semi-quantitative approach), but this represented the best option in untargeted profiling (according to literature and state-of-the-art in untargeted metabolomics). Of course, having the standard of each compound (absolute quantitation) is more rigorous, but then this would have been a targeted approach with more limited phenolic coverage. However, we plan in further works, to combine targeted (such as UHPLC-Orbitrap-MS) and untargeted (such as UHPLC-QTOF profiling) approaches in order to structurally confirm the identity of some typical compounds with a more robust level of confidence. We have added more details about the concentration levels used to build the calibration curves; please, check the revised paragraph 2.3.

Point 2.4 line 122 – what extracts? These prepared in methanol according to point 2.2? Write “ The C. oblonga methanolic extracts …”.

Authors: changed revised, accordingly.

Point 3.1 line 157 – why some abundances were equal to 1 (especially in one out of three samples), e.g., for peralgonidin 3-O-arabinoside?

Authors: We understand the author's request. Please, consider that the intensity value equal to 1 means that the compound was not found in that sample. The software attributes a value of 1, instead of zero, in order to avoid calculation problems in the following steps (i.e., calculation of fold-change distribution and statistical analysis). Regarding the example made by the reviewer for pelargonidin-3-O-arabinoside, please consider that a filter by frequency was used following the deconvolution of the different mass features, using the software Mass Profinder (Agilent Technologies). In particular, a compound was retained if found in at least the 75% of the replicates within at least one sample group.

Point 3.1 line 158 – replace trough with through.

Authors: revised, accordingly.

Point 3.1 line 164 – write MS/MS.

Authors: revised, accordingly.

Point 3.1 line 167 – the names of extracts in Table S2 give no information. Which extract is from leaf, stem, and fruit? This information is given only in Table S1.

Authors: revised, accordingly. Thank you for pointing it out.

Figure 1 Give “Whole fruit” as the first bar, then “Leaf” and “Stem” (like in Figure 2). Is it the methanolic extract according to point 2.2? The extract in the UHPLC method is not methanolic (only 80% methanol was used).

Authors: Thank you for your suggestion, we changed the figure accordingly. In addition, we have revised the extract name for the UHPLC analysis (hydroalcoholic extract).

Point 3.1 line 180 – what is appropriate – “higher in leaves and stems than in fruit” or “higher in leaves than in stems”? I cannot verify based on the information given in Tables S1 and S2.

Authors: Thank you for your suggestion. In order to better understand the data, we have added two more columns in table S3, including the values of total phenolics content and total flavonoids content.

Figure 2 – write Cydonia oblonga in italics.

Authors: revised, accordingly.

Table 1 – it should be preferably on one page – the header cannot be easily followed when it is on a separate page.

Authors: revised, accordingly.

Point 3.3 lines 286-287 – based on the information given in Table 3 there is no statistically valid difference for AChE results (all plant parts) and between BChE (for fruit and stem). You cannot write in the text that these results were higher for the stem. Further discussion in the text should also be corrected.

Authors: We agree with the reviewer and we have revised the discussion of the results. Sorry for the misleading information provided in the previous version of the manuscript.

Point 3.3 lines 302-303 – based on the information given in Table 3 there is no statistically valid difference for inhibition of amylase.

Authors: We agree with the reviewer and we have revised the discussion of the results. Sorry for the misleading information provided in the previous version of the manuscript.

Point 3.3 lines 302-305 – this is not clear – shouldn’t you write it in one sentence?

Authors: revised, accordingly.

Point 3.3 line 329 – Figure 1?

Authors: Revised, accordingly. We have now indicated Table S3, in which the semi-quantitative values for each sub-class of polyphenols have been reported for the different parts of the plant.

Reviewer 2 Report

This is a well-designed study; however, the authors are invited to provide representative chromatograms from the test samples and to revise the chemical composition in Supp Table 1. The identity of some compounds needs to be checked, for example (+)-Catechin / (-)-Epicatechin, Phloridzin.

Author Response

Reviewer #2

This is a well-designed study; however, the authors are invited to provide representative chromatograms from the test samples and to revise the chemical composition in Supp Table 1. The identity of some compounds needs to be checked, for example (+)-Catechin / (-)-Epicatechin, Phloridzin.

Authors: We would like to thank the reviewer for having appreciated this work. We fully understand the reviewer's request. We have now provided representative chromatographic results of some phenolics characterizing the different samples under investigation. Also, we have included representative total ion current chromatograms as resulting by the full-scan MS (i.e., untargeted metabolomics-based approach). This material is now available in Table S1. Regarding the need to check the identity of some compounds, please consider that as annotation strategy adopted in this work, we used a Level 2 of confidence typical of untargeted metabolomics experiments. In this regard, to the best of our knowledge, no comprehensive databases for polyphenols are available as MS/MS spectra (at least, for comprehensive databases – some commonly known databases such as FoodDB do have some of these compounds, but their number is much lower than comprehensive databases such as Phenol- Explorer) we decided to use high-resolution MS-only data to putatively annotate the compounds according to a LEVEL 2 of identification (COSMOS Standards Metabolomics Initiative), followed by a data-dependent Auto MSMS approach (based on Quality Control samples) to confirm the structural identity of some metabolites (by means of MS-DIAL and MS-FINDER workflows). Overall, as a first step, raw mass features were deconvoluted from the total ion current, and then processed by using several filters (i.e., frequency, retention time alignment, mass spectrum alignment) prior to be elaborated by means of multivariate statistics. Compound’s annotation was based on monoisotopic mass, isotopic ratio, and isotopic spacing, considering the quasi-molecular ion and the possible source adducts. We already used this approach in more than 60 published papers on high-impact journals, such as Food Chemistry, Industrial Crops and Products, Foods, Antioxidants, and Food Research International. Also, rather than using the sole pseudomolecular ion, we took advantage from high-resolution mass spectrometry and used the whole isotopic profile. In more detail monoisotopic accurate masses have been used to identify proton adducts, Na-adducts, neutral losses, electron loss, together with all their combinations. The information was combined to achieve an isotopic pattern (monoisotopic masses, isotopes spacing and isotopes ratio) for each possible adduct. The annotation process computationally scores these parameters to annotate compounds. All this information is now provided in supplementary material (revised Table S1). Overall, we were in the position of deciding between using the strongest ID confidence (MS/MS) with a limited compound coverage, rather than having a lower ID confidence but a very comprehensive profiling. We decided for the second mainly for two reasons: a) less characterized and commercially unavailable compounds (e.g., diversely conjugated and glycosylated) are often very informative and surely novel (above all when considered the matrix under investigation, i.e., quince; b) we added several ID criteria, additionally to monoisotopic accurate mass, to compensate for the lack of MS2 (namely, isotopic spacing, isotopic ratio, filters by frequency across replications, retention time alignment. We felt that the ID approach used is the strongest possible among MS-only acquisitions.

Reviewer 3 Report

The manuscript titled “The UHPLC-QTOF-MS phenolic profiling and activity of Cydonia oblonga Mill. reveals a promising nutraceutical potential” presents the chemical profile as well as the biological properties of Cydonia oblonga Mill. fruits, leaves, and stems. The authors have used UHPLC coupled to a Q-ToF-MS for the detection and identification of phenols. The MS data was interpreted using Agilent Profinder software and phenol explorer database. Further confirmation was performed using the MS-DIAL software using a number of libraries. The results were well presented in a heat-map. Correlation between phytochemical profiles and biological activities was also presented.

It is a novel study since there is no previous work on the nutraceutical profile of the different parts of Cydonia oblonga. Food industries focus on the use of the fruits part to produce jams and jellies, however this study has demonstrated that stems and leaves are also a rich source of phenolic compounds and the stems showed higher inhibitory of both AChe and BChe comparing to other parts of the plant.

Overall well written manuscript, however, some parts are hard to read especially with the long sentences, and long enumerations of phytochemicals compounds.

Below some comments:

Line 47. Correct the sentence “In addition to fruits, 46 Cydonia leaves have been studied because of their important constituents…”

Line 55. Remove the word “untargeted”

Line 82. Replace “MS-only” by “MS full scan”

Line 93 to 103. Very long sentence please cut it

Line 158. Correct the sentence “Also, using the tandem-mass spectrometry approach, the structural identity of some phytochemicals was confirmed”

Line 159-165. Very long sentence please cut it. It is hard to read.

Line 160. Replace “confirm” with “detect and confirm”

Line 177. Correct the sentence. The flavonoid content in leaves was almost double the amount found in stems.

Figure 2. Interesting figure of the heat map. Could you describe more the column on the left in the figure? Does it correspond to the peak area of phytochemicals found? The cluster of leaves and stems is not exactly the same; you should mention that somehow in the manuscript, especially looking at the top of the heat map, we see some differences in the colors.

Line 230. Could you add how the VIP score was calculated?

Line 294-298. This paragraph should be moved to paragraph 3.1.

L 360. Replace “poses it at” by “makes it” or rephrase the sentence.

Author Response

Reviewer #3

The manuscript titled “The UHPLC-QTOF-MS phenolic profiling and activity of Cydonia oblonga Mill. reveals a promising nutraceutical potential” presents the chemical profile as well as the biological properties of Cydonia oblonga Mill. fruits, leaves, and stems. The authors have used UHPLC coupled to a Q-ToF-MS for the detection and identification of phenols. The MS data was interpreted using Agilent Profinder software and phenol explorer database. Further confirmation was performed using the MS-DIAL software using a number of libraries. The results were well presented in a heat-map. Correlation between phytochemical profiles and biological activities was also presented.

It is a novel study since there is no previous work on the nutraceutical profile of the different parts of Cydonia oblonga. Food industries focus on the use of the fruits part to produce jams and jellies, however this study has demonstrated that stems and leaves are also a rich source of phenolic compounds and the stems showed higher inhibitory of both AChe and BChe comparing to other parts of the plant.

Overall well written manuscript, however, some parts are hard to read especially with the long sentences, and long enumerations of phytochemicals compounds.

Authors: We would like to thank the reviewer for having appreciated this work. A deep revision has been done according to the major drawbacks raised by the reviewer.

Below some comments:

Line 47. Correct the sentence “In addition to fruits, 46 Cydonia leaves have been studied because of their important constituents…”

Authors: revised, accordingly.

Line 55. Remove the word “untargeted”

Authors: revised, accordingly.

Line 82. Replace “MS-only” by “MS full scan”

Authors: revised, accordingly.

Line 93 to 103. Very long sentence please cut it

Authors: revised, accordingly.

Line 158. Correct the sentence “Also, using the tandem-mass spectrometry approach, the structural identity of some phytochemicals was confirmed”

Authors: revised, accordingly.

Line 159-165. Very long sentence please cut it. It is hard to read.

Authors: revised, accordingly.

Line 160. Replace “confirm” with “detect and confirm”

Authors: revised, accordingly.

Line 177. Correct the sentence. The flavonoid content in leaves was almost double the amount found in stems.

Authors: revised, accordingly.

Figure 2. Interesting figure of the heat map. Could you describe more the column on the left in the figure? Does it correspond to the peak area of phytochemicals found? The cluster of leaves and stems is not exactly the same; you should mention that somehow in the manuscript, especially looking at the top of the heat map, we see some differences in the colors.

Authors: We would like to thank the reviewer for the question. Indeed, Figure 2 corresponds to the heat map created by the hierarchical unsupervised cluster analysis, using the software Mass Profiler Professional (by Agilent Technologies). The column on the left represents all the compounds annotated (in particular, each row corresponds to a single compound), while on the top of the figure we can observe the unsupervised grouping of the different sample replicates according to the phytochemical profile detected. In particular, the heat map is built considering the fold-change variation of each compound across the different sample replicates. The color range in each column represents the maximum (red) and minimum (blue) values of fold-change variations, while the yellow color means no significant variation. The variability in the "color-distribution" observed (as suggested by the reviewer) is due to the analysis of n=3 biological replications for each sample (i.e., leaf, stem, and fruit). Finally, it is important to mention that the fold-change-based heat map is provided taking into account the raw peak areas of each compound annotated by UHPLC-QTOF mass spectrometry.

Line 230. Could you add how the VIP score was calculated?

Authors: The VIP score resulting from the variables importance in projection following the OPLS-DA prediction model, is built by the software (SIMCA, Umetrics) and it displays the overall importance of each variable (x) on all responses (Y) cumulatively over all components. In particular, interpreting an OPLS model with many components and a multitude of responses can be a complex task. A parameter which summarizes the importance of the X-variables, both for the X- and Y-models, is called the variable influence on projection, VIP. For PLS and OPLS model, VIP is a weighted sum of squares of the PLS weights, w*, taking into account the amount of explained Y-variance in each dimension. Its attraction lies in its intrinsic parsimony; for a given model and problem there will always be only one VIP-vector, summarizing all components and Y-variables. One can compare the VIP of one term to the others. Terms with large VIP, larger than 1, are the most relevant for explaining Y. The VIP values reflect the importance of terms in the model both with respect to Y, i.e., its correlation to all the responses, and with respect to X (the projection). With designed data, i.e., close to orthogonal X, the VIP values mainly reflect the correlation of the terms to all the responses. VIP values are computed, by default, from all extracted components. To take advantage of the interpretational clarity of OPLS, VIP for OPLS consists of three vectors, VIP for the predictive components (VIP predictive), VIP for the orthogonal components (VIP orthogonal), and VIP for the total model (VIP total). In each one of these three vectors, the VIP values are regularized such that if all X-variables would have the same importance for the model they would all have the value 1. Consequently, terms with VIP values larger than 1 in either VIP total, VIP predictive or VIP orthogonal, point to variables with large importance for that part of the model. We have added the description of the calculation of VIP score in the revised manuscript.

Line 294-298. This paragraph should be moved to paragraph 3.1.

Authors: We would like to thank the reviewer for the suggestion. However, we preferred to keep the description on lignans in that part of the discussion, considering that we cited lignans as related to inhibition of AChE and BChE enzymes.

L 360. Replace “poses it at” by “makes it” or rephrase the sentence.

Authors: revised, accordingly.

Round 2

Reviewer 1 Report

The manuscript “The UHPLC-QTOF-MS phenolic profiling and activity of Cydonia oblonga Mill. reveals a promising nutraceutical potential” in its second version contains many changes and is better readable. It needs only two small corrections before publication.

Details:

Lines 241-242 Remove one “overall”, e.g. write “The VIP scores displays the overall importance of each variable”

Table S3 A few columns should be wider to avoid values and superscripts being in two separate lines.

Author Response

The manuscript “The UHPLC-QTOF-MS phenolic profiling and activity of Cydonia oblonga Mill. reveals a promising nutraceutical potential” in its second version contains many changes and is better readable. It needs only two small corrections before publication.

Authors: We would like to thank the reviewer for having appreciated this work. The minor changes have been carefully considered in the revised version of the manuscript.

Details:

Lines 241-242 Remove one “overall”, e.g. write “The VIP scores displays the overall importance of each variable”

Authors: revised, accordingly.

Table S3 A few columns should be wider to avoid values and superscripts being in two separate lines.

Authors: Table S3 has been revised, accordingly. Thank you for pointing it out.